# High-Altitude Exposure and Time Interval Perception of Chinese Migrants in Tibet

**DOI:** 10.3390/brainsci12050585

**Published:** 2022-04-29

**Authors:** Yuan Li, Mei-Yi Wang, Meng Xu, Wen-Ting Xie, Yu-Ming Zhang, Xi-Yue Yang, Zhi-Xin Wang, Rui Song, Liu Yang, Jin-Ping Ma, Jia Zhang, Chen-Xiao Han, Cheng-Zhi Wang, Wan-Ying Liu, Wan-Hong Gan, Rui Su, Hai-Lin Ma, Hao Li

**Affiliations:** 1Plateau Brain Science Research Center, Tibet University/South China Normal University, Lhasa 850012, China; charleslilovepeace@163.com (Y.L.); wyiyi412@163.com (M.-Y.W.); xm145690@163.com (M.X.); gszyxwt@126.com (W.-T.X.); minging771@163.com (Y.-M.Z.); xiyue19980912@163.com (X.-Y.Y.); wangzhixin666@outlook.com (Z.-X.W.); segniusyl@126.com (L.Y.); m9958154341@163.com (J.-P.M.); 15629345581@163.com (J.Z.); cxiao200601@163.com (C.-X.H.); wcz9803@163.com (C.-Z.W.); liuwypsy@163.com (W.-Y.L.); gwh_456@163.com (W.-H.G.); srsisu2011@163.com (R.S.); david_ma79@163.com (H.-L.M.); 2Center on Aging Psychology, CAS Key Laboratory of Mental Health, Institute of Psychology, Beijing 100101, China; 18202265323@163.com; 3Institute of Oxygen Supply, Tibet University, Lhasa 850012, China

**Keywords:** high altitude, time perception, time interval judgment, sleep quality, sleepiness

## Abstract

High-altitude exposure can negatively impact one’s ability to accurately perceive time. This study focuses on Chinese migrants who have traveled to the Tibetan plateau and explores the effects of high-altitude exposure on their time interval judgment abilities based on three separate studies. In Study 1, it was found that exposure to high altitudes negatively impacted the time interval judgment functions of the migrants compared with a low-altitude control group; they exhibited a prolonged response time (540 ms: *p =* 0.006, 95% *CI* (−1.70 −0.32)) and reduced accuracy (1080 ms: *p* = 0.032, 95% *CI* (0.06 1.26)) in certain behavioral tasks. In Study 2, the results showed that high-altitude exposure and sleepiness had an interactive effect on time interval judgment (1080 ms) (*p* < 0.05, 95% *CI* (−0.83 −0.40)). To further verify our interaction hypothesis, in Study 3, we investigated the time interval judgment of interactions between acute high-altitude exposure and sleepiness level. The results revealed that the adaptation effect disappeared and sleepiness significantly exacerbated the negative effects of high-altitude exposure on time interval judgment (*p <* 0.001, 95% *CI* (−0.85 −0.34)). This study is the first to examine the effects of high-altitude exposure on time interval judgment processing functions and the effects of sleep-related factors on individual time interval judgment.

## 1. Introduction

Time perception is an individual’s reflection on the continuity and sequence of objective events that directly affect multiple sensory organs simultaneously [1]. In China, Bao et al., (2013) were the first to explore the effects of acute high-altitude hypoxic exposure on time perception in migrants [2]. They found that the time perception performance of those living in the Chinese plain was better than that of those who occupied the high elevations of the Tibetan plateau, and there was a statistically significant difference between hypoxia levels before and after acute high-altitude exposure. Time perception includes the understanding of interval timing, time sequence, and time points; however, in the existing research, the focus has mainly been on interval timing [3]. Time perception is essential for one’s survival, in which time perceptions ranging from hundreds of milliseconds to several minutes highly influence various behaviors in life [4]. Time perception within this range is called interval timing. In previous studies, Yin, Huang, and Daniels et al. studied interval timing; however, high-altitude hypoxia from a specific high-elevation environment was not considered in their studies [5,6,7].

In high-altitude areas, oxygen particles are further apart because there is less pressure in the air to “push” them together. Thus, when air is inhaled, individuals experience insufficient oxygenation in their blood and lungs; this is one of many effects that a plateau environment has on the human body [8]. Although the brain makes up only about 2% of the body’s weight, it consumes 20% of its oxygen intake [9,10]. This undoubtedly leads to an insufficient oxygen supply in the brains of residents living in high-altitude areas. Long-term and repeated exposure to high-altitude environments can lead to central nervous system damage and affect cognitive function [11,12]. Despite adaptive evolution, maladaptive changes in neurological and cognitive function have been observed in native populations, particularly in long-term settlers [13]. This restricts the work efficiency of people living on plateaus, especially those with jobs that require high cognitive resources, such as scientific research and business management.

Previous studies have shown that hypoxia due to long-term high-altitude exposure can affect a series of basic cognitive functions, such as attention and working memory, for non-adaptive people (plateau migrants) [14,15,16,17,18]. Long-term or chronic hypoxic exposure mainly affects prefrontal lobe function [19]. At the same time, it has been proven that, with an increase in altitude, the functions of brain areas sensitive to hypoxia, such as the hippocampus, striatum, and cerebellum, significantly decrease when hypoxic [20]. Long-term exposure to high-altitude and low-oxygen environments can lead to hypoxia and thus decreased cognitive function. In the case of Chinese migrants who have not yet adapted to the environment of the Tibetan plateau, issues such as visual, hearing, and spatial memory [19,20] problems are common. The brain regions associated with these functions are closely associated with time perception—e.g., the prefrontal lobe has functions in long-term time estimation, while the cerebellum is associated with short-term time perception [21]. Based on this, we propose Hypothesis 1 as follows: Exposure to high altitudes (3500 ± 200 m) for more than 1 year (≥1 year) has a negative effect on the time interval judgment of Chinese migrants in the Tibetan plateau.

Some studies have pointed out that compromised sleep quality, which is a common physiological effect of high altitudes, is a serious threat to the physical and mental health of plateau migrants [22,23,24]. The World Health Organization conducted a survey that shows that more than 27% of the world’s people experience sleep problems. Meanwhile, plateau life has a greater negative impact on people’s sleep quality, causing declines in sleep quality as well as changes in sleep structure. As a result of these issues, individuals may experience an increase in sleepiness and even physical and mental health problems [25,26,27,28]. According to the arousal hypothesis, sleep restriction interferes with task performance largely due to the significantly reduced physiological arousal level of individuals [29]. When individuals’ sleep duration is limited, their arousal level will be too low to achieve optimal task performance; therefore, their task performance deteriorates [30]. Meanwhile, the interval timing processing of time perception is not only related to the cognitive control process but also to arousal factors [21]. Sleep loss leads to a significant increase in subjective sleepiness and fatigue levels and eventually to a significant decrease in individual inhibitory control abilities [31]. Sleepiness may also affect the time interval judgment of individuals.

As discussed above, several studies have shown that sleep deprivation and decreased sleep quality can lead to the compromised functioning of various cognitive processes, including the ability to accurately perceive time intervals [25,26,27,28]. Additionally, individuals exposed to high-altitude and low-oxygen environments for a prolonged period will suffer from long-term hypoxia due to insufficient blood oxygen levels [8], impacting regions such as the prefrontal lobe, cerebellum, and hypothalamus [19,20], which are highly correlated with time interval judgment abilities [32]. Factors associated with sleep (such as the effect of sleepiness) also strongly influence an individual’s time interval judgment abilities and time perception. We thus propose the second hypothesis as follows: Different altitude levels (plateau vs. plain areas) and individual sleepiness levels interact to affect individual time interval judgments. We expect that exposure to high-altitude and low-oxygen environments may negatively affect time interval judgment performance at the same degree of sleepiness.

Worldwide, more than 140 million people live at altitudes above 2500 m, mainly in the Americas, East Africa, and Asia [33]. The high-altitude and low-oxygen environment affect the health, work, and life of settlers who have migrated from the plains of China to the Tibetan plateau, also known as “the roof of the world”. To find evidence of the long-term influence of high-altitude exposure on time interval judgment, this study provides a theoretical basis for the adaptation of plateau migrants and expands the current direction of plateau-related brain science research. The following three studies were conducted. The first study investigated the effects of exposure to high-altitude and low-oxygen environments and the time interval judgment abilities of migrant settlers. In the second study, we investigated whether prolonged exposure to a high-altitude environment had a greater negative impact on the time interval judgment abilities of sleep-deprived individuals. In the third study, we further evaluated the interactive effects of acute high-altitude exposure and sleepiness on time interval judgment ability. Our study provides a theoretical basis for the importance of sleep and the cognitive processes of time perception in highland residents around the world. Whether various factors related to sleep will affect individuals’ time interval judgment abilities also presents implications for the next steps of this research.

## 2. Study 1

### 2.1. Participants

We enrolled a total of 60 participants. In this study, 30 people were enrolled in the plateau migration and plain control groups, respectively. In addition, the sample size was reasonable, which was measured G*Power 3.1 [34] (*t =* 2.00, effect size was 0.95). All participants were randomly selected from Tibet University and universities in Tianjin. At the same time, there were no significant differences in sex and age between the two groups (*χ*^2^ = 0.27, *p* > 0.05, 95% *CI* (0.28 2.11); *t* = 1.69, *p* > 0.05, 95% *CI* (−0.08 0.74)).

In the high-altitude migration group, there were 30 participants, including 15 males and 15 females, originally born in low-altitude areas below 1000 m. These participants had no prior experience living in a high-altitude area before entering Tibet. These participants were enrolled in Tibet University and ranged in age from 23 to 31 years, with an average age of 25.42 ± 1.61 years. They had been living in Lhasa, Tibet’s capital (3650 m above sea level), for approximately 1 to 2 years. The average altitude of their original residence was 495.92 m before they came to Tibet. Participants reported that they rarely left Lhasa for long durations, except for the summer and winter holidays.

The plain control group consisted of 30 people, and included 15 males and 15 females who were born in low-altitude areas below 1000 m and had no prior experience living in high-altitude areas. Participants ranged in age from 21 to 25 years old, with an average age of 22.08 ± 0.64 years. The average altitude of their long-term residence was 275 m.

Candidates were excluded from the study if they presented with severe mental disorders such as schizophrenia, affective disorder, developmental delay, or major physical and neurological diseases, or if they had been prescribed psychiatric medication. This experiment was approved by the Ethics Committee of Tibet University (XZTU2021ZRG-05). All subjects signed informed consent before the experiment and were compensated for their participation.

### 2.2. Experimental Design

Using a single-factor between-subjects design, a total of 60 subjects were randomly selected from Tibet University and universities in Tianjin and included in the plateau migration group and plain control group, respectively. Under the condition that there was no significant difference in age and gender between the two groups, the PANAS scale was firstly used to measure the level of positive and negative emotions in each group, and then, the two groups were tested, respectively, using two time interval discrimination paradigms. Finally, under the condition that there was no significant difference between the positive and negative emotion levels of the two groups, the time interval judgment levels of the two groups were compared.

### 2.3. Materials

To control for the influence of emotion on the time interval judgment abilities of the two groups, all participants were tested with the Positive and Negative Affect Schedule (PANAS) before the behavioral experiment. This measurement tool was revised in Chinese by Zhang et al., (2001) based on the original Positive and Negative Affect Schedule (PANAS) compiled by Washton et al. [35]. This scale consists of 19 items divided into two dimensions: positive emotions (9 items) and negative emotions (10 items). On a scale of 1–5, with 1 being little or none and 5 being extremely, participants were asked to describe their emotional experiences over several weeks. The average score for the items was used to indicate the frequency of positive and negative emotions. Previous research has shown that this scale exhibits good reliability and validity. The Cronbach’s α coefficient of the positive emotion subscale was 0.795, and the Cronbach’s α coefficient of the negative emotion subscale was 0.872 [36].

### 2.4. Procedures

This study adopted the “single stimulus method” and the “synchronous method” in the time interval discrimination paradigm [37]. The “synchronous method” required subjects to make synchronous replications according to the fixed time interval standard presented, while the “single stimulus method” required subjects to make long or short judgments about the time interval presented one by one. All programs were processed with e-Prime 3.0 and presented with experimental stimulation using a 14-inch laptop.

The experimental procedure of the “single stimulus method” is shown in Figure 1. All of the participants practiced several times before the formal experiment began. A 540 ms/1080 ms/1620 ms black “●” (standard stimulus S1) was randomly presented to the subjects in the middle of the screen. After an interval of 1500 ms, the subject was presented with a 540 ms/1080 ms/1620 ms black “▲” (target stimulus S2). Following this, the subjects were asked to judge whether the standard stimulus S1 was equal to the target stimulus S2, as well as whether the presentation time of S2 was shorter, equal to, or longer than that of S1. The index finger, middle finger, and ring finger were used to press one of three number keys “1”, “2”, and “3” (“1” meant shorter; “2” indicated equality; “3” meant longer). On the left hand, the ring finger pressed “1”, the middle finger pressed “2”, and the index finger pressed “3”. On the right hand, the index finger pressed “1”, the middle finger pressed “2”, and the ring finger pressed “3”. Left-handed people were asked to try and react with their left hand and right-handed people were asked to try and react with their right hand. The experiment consisted of two parts: an exercise stage consisting of three trials and a formal experiment consisting of 90 trials. Each stimulus type was randomly presented. The experiment lasted for approximately 6 min.

The experimental procedure of the “synchronization method” is shown in Figure 2. Here, all subjects practiced several times before the formal experiment. In the middle of the screen, the subjects were randomly presented a 540/1080/1620 ms text or a 2160 ms black “low” text (standard stimulus S1), followed by an interval of 300 ms. Then, the subjects observed a black “bring” text (target stimulus S2). Following this, the subjects were required to perform a synchronous replication according to the fixed time interval standard of the presented stimulus. When subjects subjectively felt that the presentation time of the target stimulus was the same as that of the standard stimulus, they pressed the space bar. This experiment consisted of an exercise phase consisting of four trials, followed by a formal experiment with 60 trials. Each stimulus type was randomly presented. The experiment lasted approximately 6 min.

### 2.5. Statistical Analysis

The SPSS 26.0 software was used for statistical analysis. All data were standardized before formal data processing. First, the independent sample *T* test was used to investigate whether there was a significant difference in PANAS score between the two groups—that is, whether there was a significant difference in positive and negative emotions between the two groups. Then, the independent sample *T* test was used to compare the behavioral differences between the two groups in two time interval discrimination paradigms.

### 2.6. Results

Before analyzing behavioral data, according to the positive–negative emotion scale and using the independent sample *T* test, we first analyzed the positive and negative emotion scores of the two groups of subjects. The results showed that there was no significant difference between the two groups of subjects in both positive and negative emotion scores (*t* = 1.95, *p* > 0.05, 95% *CI* (−0.26 8.43); *t* = 0.34, *p* > 0.05, 95% *CI* (−6.86 9.53)).

The results of the single stimulus method were as follows:

The total accuracy of time interval judgment in the migration group was significantly lower than that in the plain group (*t* = 3.35, *p* = 0.003, 95% *CI* (0.25 1.04)), and the total response time in the migration group was significantly longer than that in the plain group (*t* = −2.43, *p* = 0.024, 95% *CI* (−0.91 −0.07)). The accuracy of time interval judgment (540 ms) in the migration group was significantly lower than that in the plain group (*t* = 1.93, *p* = 0.047, 95% *CI* (0.03 0.76)), and the response time of the time interval judgment (540 ms) in the migration group was significantly longer than that in the plain group (*t* = −3.04, *p* = 0.006, 95% *CI* (−1.70 −0.32)). The accuracy of time interval judgment (1080 ms) in the migration group was significantly lower than that in the plain group (*t* = 2.29, *p* = 0.032, 95% *CI* (0.06 1.26)). The accuracy of the time interval judgment (1620 ms) in the migration group was significantly lower than that in the plain group (*t* = 2.62, *p* = 0.016, 95% *CI* (0.18 1.54)). All of the above results are shown in Figure 3 and Figure 4.

The results of the synchronization method were as follows:

The time interval replication of the 2160 ms duration of the standard stimulus was significantly longer in the migration group than in the plain group (*t* = −2.08, *p* = 0.043, 95% *CI* (−1.114 −0.001)). For other standard stimuli, no statistical difference was found in the time interval replication between the two groups. The results are shown in Figure 5.

## 3. Study 2

### 3.1. Participants

A minimum of 54 participants were required for this study, as measured using G*Power 3.1 [34]. Based on this requirement, we recruited another 56 participants, including 28 in the plateau migration group and 28 in the plain control group, who were required to be consistent with Experiment 1. All of the participants were randomly recruited from Tibet University and universities in Beijing. There were also no significant differences in sex or age between the two groups (*χ*^2^
*<* 0.001, *p* > 0.05, 95% *CI* (0.35 2.86); *t* = 1.06, *p* > 0.05, 95% *CI* (−0.39 1.24)).

In the plateau migration group, there were 14 men and 14 women ranging in age from 23 to 31 years old, with an average age of 25.93 ± 2.22 years. These participants lived in Lhasa (3680 m above sea level) for one year, and the average altitude of their home regions, before entering Tibet, was 424.76 m. All participants were born in low-altitude areas below 1000 m and had no prior experience living in high-altitude areas before entering Tibet. The participants lived in Lhasa for 1–2 years and reported never leaving Lhasa for long durations except for the summer and winter holidays.

In the plain control group, there were 14 men and 14 women ranging in age from 21 to 25 years old, with an average age of 22.38 ± 1.04 years. They had never lived in high-altitude areas, and the average altitude of their long-term residence was 43.5 m. All participants were born at altitudes below 1000 m and had no previous experience of living at high altitudes.

The control conditions of all subjects in Study 2 were consistent with those of Study 1. This experiment was approved by the Ethics Committee of Tibet University (XZTU2021ZRG-05). All subjects signed informed consent before the experiment and were compensated for their participation.

### 3.2. Experimental Design

An intergroup (group) intra group (sleepiness level) mixed design of 2 (group: plateau migration group, plain control group) *2 (high sleepiness, low sleepiness) was used. A total of 56 subjects were randomly selected from Tibet University and universities in Beijing and included in the plateau migration group and the plain control group. The scores on the PANAS scale were measured for the two groups to ensure that there was no significant difference in emotional level between the two groups. The languid/vigorous subscales of the Chinese version of CTI-11 were used to measure the sleepiness levels of the two groups. Meanwhile, the time interval judgment levels of the two groups were measured using two time interval discrimination paradigms. Finally, the effect of sleepiness level on the performance of time interval judgment was examined to explore whether there was a significant interaction between sleepiness level and high-altitude exposure on the performance of time interval judgment.

### 3.3. Materials

To control for the influence of emotion on the time interval judgment abilities of the two groups, all participants were tested with the Positive and Negative Affect Schedule (PANAS) before the behavioral experiment, as in Study 1. The Cronbach’s α coefficient of the positive emotion subscale was 0.82, and the Cronbach’s α coefficient of the negative emotion subscale was 0.83 [36].

The Chinese version of the CTI-11 was used [38]. In this study, the languid/vigorous subscale (6 items, e.g., “If I get up early in the morning, I’ll feel tired all day”) was used to measure the participants’ sleepiness levels. It has shown good reliability and validity in previous studies. The Cronbach’s α coefficient was 0.74. A Likert 5-level scoring method was used to assign 1 to 5 points, with 5 being extremely inconsistent and 1 being extremely consistent. The scores for the languidness/vigorousness dimension ranged from 6 to 30 points. The higher the score was, the lower the vitality level of the subjects was and the weaker their ability to overcome sleepiness was.

### 3.4. Procedures

Consistent with Study 1, Study 2 adopted the “single stimulus method” and “synchronous method” for the time distance discrimination paradigm [37]. All of the programs were created using e-Prime 3.0 and presented with experimental stimulation using a 14-inch laptop.

### 3.5. Statistical Analysis

The SPSS 26.0 software was used for statistical analysis. All data were standardized before formal data processing. Firstly, a correlation analysis of the sleepiness level, 1080 ms accuracy, and group was carried out. Secondly, the accuracy at 1080 ms and the sleepiness level of the two groups were compared using an independent sample *T* test. Finally, in order to test the influence of sleepiness level on the accuracy of time interval judgment (1080 ms) and the moderating effect of group, the PROCESS Model 1 developed by Hayes [39] was used to test the moderating effect.

### 3.6. Results

Similarly to Study 1, an independent sample *T* test was conducted on the emotional scores of the two groups of participants before analyzing the behavioral experimental data. The results revealed that there was no significant difference between the two groups of subjects in both positive and negative emotional scores (*t* = 1.69, *p* > 0.05, 95% *CI* (−0.30 0.97); *t* = 0.78, *p* > 0.05, 95% *CI* (−0.26 0.59)).

In Study 1, it was found that the total accuracy of time interval judgment ability, the accuracy of time interval judgment of 540 ms, and the accuracy of time interval judgment of 1080 ms in the plain group were significantly higher than in the migration group. Therefore, the effects of sleepiness level and different altitude exposure levels on time interval judgment abilities were investigated under the same control conditions as those in Study 1.

#### 3.6.1. Descriptive Statistics

Table 1 exhibits how there was no significant correlation between languid/vigorous levels and the correct time interval judgment (1080 ms); however, there was a significant positive correlation within the group. The results of the descriptive statistics and correlation analyses are shown in Table 1.

The independent sample *T* test showed that there was no significant difference in the accuracy of time interval judgment (1080 ms) between the plateau migration group (*M* = 0.65, *SD* = 0.09) and the plain control group (*M* = 0.67, *SD* = 0.13; *t* = −0.49, *p* > 0.05, 95% *CI* (−0.54 0.26)); the scores for languid/vigorous levels in the plain control group (*M* = 20.77, *SD* = 3.90) were significantly higher than those in the plateau migration group (*M* = 17.53, *SD* = 4.31; *t* = −2.07, *p* < 0.05, 95% *CI* (−1.25 −0.25)).

#### 3.6.2. Moderating Effect Test

To test the effect of languid/vigorous levels on the accuracy of time interval judgment ability (1080 ms) and the moderating effect of the groups, PROCESS Model 1, developed by Hayes (2013) [39], was used to test the moderating effects (see Table 2). The results revealed that the direct prediction effect of languid/vigorous levels on the accuracy of time interval judgment (1080 ms) was not significant (*β* = 0.04, *t* = 0.87, *p* > 0.05, 95% *CI* (−0.11 0.26)), and the prediction effect of groups was not significant (*β* = 0.32, *t* = 1.13, *p* > 0.05, 95% *CI* (−0.10 0.38)). However, the product of languid/vigorous levels among the groups had a significant predictive effect (*β* = −0.17, *t* = −2.58, *p* < 0.05, 95% *CI* (−0.83 −0.40)).

As shown in Table 3 and Figure 6, the accuracy of time interval judgment abilities (1080 ms) of college students in the plain control group was lower than the scores for languid/vigorous levels, and these levels were higher (*β* = −0.19, *t* = −2.75, *p* < 0.05, 95% *CI* (−0.87 −0.28)). In summary, the accuracy of time interval judgment (1080 ms) decreased with sleepiness or lack of sleep, and the correct rate of time interval judgment (1080 ms) did not change with the languid/vigorous levels (*β* = 0.11, *t* = 1.73, *p* > 0.05, 95% *CI* (−0.09 0.41)).

## 4. Study 3

While the second study measured the accuracy of the time interval judgment (1080 ms) of the plateau group, which was higher than that of the plain control group under a high level of sleepiness, the results were the opposite under a low level of sleepiness, indicating that sleep deprivation adaptation occurred in the plateau group due to long-term high-altitude exposure. To further verify Hypothesis 2, Study 3 discussed the interactive effects of sleep quality and altitude on immigrant time interval judgment abilities under acute high-altitude exposure (within 1 week).

### 4.1. Participants

At least 54 participants were required for this study using the G*Power 3.1 software [34], with 40 subjects in the acute exposure group and 30 subjects in the plain control group in the formal experiment, who were required to be consistent with Study 1. All participants were also randomly recruited again from Tibet University and universities in Beijing. Among them, participants from Tibet University were included in the acute exposure group. They were freshmen who had just arrived in Lhasa for less than a week. After excluding outliers, there were 33 participants in the acute exposure group and 27 in the plain control group. There were also no significant differences in sex or age between the two groups (*χ*^2^ = 0.93, *p* > 0.05, 95% *CI* (0.22 1.69); *t* = 1.13, *p* > 0.05, 95% *CI* (−0.19 0.64)).

In the acute exposure group, there were 13 men and 20 women ranging in age from 21 to 29 years old, with an average age of 23.85 ± 1.99 years. These participants arrived in Lhasa (3680 m above sea level) within 1 week. All participants were born in low-altitude areas below 1000 m and had no prior experience living in high-altitude areas before entering Tibet.

In the plain control group, there were 14 men and 13 women ranging in age from 21 to 23 years old, with an average age of 22.11 ± 0.64 years. They had never lived in high-altitude areas. All participants were born at altitudes below 1000 m and had no previous experience with living at high altitudes.

The control conditions of all subjects in Study 3 were consistent with those of Study 1. This study was approved by the Ethics Committee of Tibet University (XZTU2021ZRG-05). All subjects signed informed consent forms before the experiment and were compensated for their participation.

### 4.2. Experimental Design

An intergroup (group) intra group (sleepiness level) mixed design of 2 (group: acute exposure group, plain control group) *2 (high sleepiness, low sleepiness) was used. Sixty subjects were randomly selected from Tibet University and universities in Beijing and included in the acute exposure group and the plain control group. The scores on the PANAS scale were measured for the two groups to ensure that there was no significant difference in emotional level between the two groups. The languid/vigorous subscale of the Chinese version of CTI-11 was used to measure the sleepiness levels of the two groups. Meanwhile, the time interval judgment performance of the two groups was measured using two time interval discrimination paradigms. Finally, the effect of sleepiness level on the performance of time interval judgment was examined to explore whether there was a significant interaction between sleepiness level and acute high-altitude exposure on the performance of time interval judgment.

### 4.3. Materials

To control for the influence of emotion on the time interval judgment abilities of the two groups, all participants were tested using the Positive and Negative Affect Schedule (PANAS) before the behavioral experiment, as in Study 2. The Cronbach’s α coefficient of the positive emotion subscale was 0.79, and the Cronbach’s α coefficient of the negative emotion subscale was 0.81 [36].

As in Study 2, the Chinese version of the CTI-11 was used [38]. The languid/vigorous subscale (6 items, e.g., “If I get up early in the morning, I’ll feel tired all day”) was used to measure participants’ sleepiness levels. The Cronbach’s α coefficient was 0.77. Consistent with Study 2, Study 3 adopted the “single stimulus method” and “synchronous method” of the time distance discrimination paradigm [37]. All programs were created using e-Prime 3.0 and presented with experimental stimulation using a 14-inch laptop.

### 4.4. Procedures

Consistent with Study 1, Study 3 adopted the “single stimulus method” and “synchronous method” of the time distance discrimination paradigm [37]. All programs were created using e-Prime 3.0 and presented with experimental stimulation using a 14-inch laptop.

### 4.5. Statistical Analysis

SPSS26.0 was used for descriptive statistical analysis, and an independent sample *T* test and PROCESS3.2 were used for the moderating effect test. All data were standardized before formal data processing was conducted. Firstly, a correlation analysis of sleepiness level, 1080 ms accuracy, and group was carried out. Secondly, the accuracy of the 1080 ms judgment and the sleepiness level of the two groups were compared using an independent sample *T* test. Finally, in order to test the influence of sleepiness level on the accuracy of time interval judgment (1080 ms) and the moderating effect of group, the PROCESS Model 1 developed by Hayes [39] was used to test the moderating effect.

### 4.6. Results

Similarly to Study 2, an independent sample *T* test was conducted on the emotional scores of the two groups of participants before analyzing the behavioral experimental data. The results revealed that there was no significant difference between the two groups of subjects in terms of both positive and negative emotional scores (*t* = 1.59, *p* > 0.05, 95% *CI* (−0.28 0.95); *t* = 0.83, *p* > 0.05, 95% *CI* (−0.22 0.63)).

The effects of sleepiness level and different altitude exposure levels (acute exposure group vs. plain control group) on participants’ time interval judgment abilities were investigated under the same control conditions as those used in Study 2.

#### 4.6.1. Descriptive Statistics

As can be seen in Table 4, there was a significant correlation between languid/vigorous level, the accuracy of time interval judgment ability (1080 ms), and the groups, with correlation coefficients of −0.37 and 0.33, respectively. The results of the descriptive statistics and correlation analyses are shown in Table 4.

In the independent sample *T* test, we found that the acute exposure group (*M* = 0.64, *SD* = 0.10) and plain control group (*M* = 0.68, *SD* = 0.11; *t* = −1.51, *p* > 0.05, 95% *CI* (−0.63 0.09)) had certain differences; however, there was no significant difference in the accuracy of time interval judgment (1080 ms), and the scores for languid/vigorous levels in the plain control group (*M* = 18.04, *SD* = 2.54) were significantly lower than those in the acute exposure group (*M* = 19.74, *SD* = 2.47; *t* = −2.61, *p* < 0.05, 95% *CI* (−0.98 −0.13)).

#### 4.6.2. Moderating Effect Test

To test the effect of languid/vigorous level on the accuracy of time interval judgment ability (1080 ms) and the moderating effect of groups (acute exposure group and plain control group), the PROCESS Model 1, developed by Hayes (2013) [39], was used to test the moderating effects (see Table 5). The results revealed that languid/vigorous level had a significant effect on the accuracy of time distance judgment (1080 ms) (*β* = −0.59, *t* = −4.67, *p* < 0.001, 95% *CI* (−0.55 −0.16)), while the predictive effect of the groups was also significant (*β* = 0.46, *t* = 2.83, *p* < 0.01, 95% *CI* (0.03 0.37)), and the product of languid/vigorous levels and the groups showed a significant predictive effect (*β* = 0.45, *t* = 2.33, *p* < 0.05, 95% *CI* (0.03 0.42)).

The difference analysis carried out among the influences of each group (acute exposure group and plain control group) on the accuracy of time interval judgment (1080 ms) showed that (shown in Table 6 and Figure 7) the accuracy of time interval judgment ability (1080 ms) decreased as the scores for languid/vigorous levels in the acute exposure group became higher and higher (*β* = −1.06, *t* = −3.53, *p* < 0.001, 95% *CI* (−0.85 −0.34))—that is, the accuracy of time interval judgment (1080 ms) decreased with sleepiness and a lack of sleep, whereas in the plain control group the accuracy of time interval judgment (1080 ms) did not change with the change in the languid/vigorous level (*β* = −0.17, *t* = −1.26, *p* > 0.05, 95% *CI* (−0.44 0.15)).

## 5. Discussion

This study is the first of its kind to preliminarily explore the effects of high-altitude exposure on the time interval judgment abilities of Chinese migrants who have moved to the Tibetan plateau. In Study 1, it was found that long-term exposure to high altitudes had a negative impact on the time interval judgment functions of migrants, mainly reflected in the prolonged response time and decreased accuracy of the high-altitude migrant group during behavioral tasks. This validates Hypothesis 1 in this study. In Study 2, altitude moderated the effect of sleepiness on time interval judgment (1080 ms) under long-term high-altitude exposure. This validates Hypothesis 2. In Study 3, we investigated the effects of altitude and sleepiness level on the time interval judgment abilities of settlers under acute high-altitude exposure and validated the results of Study 2. It is, therefore, suggested that there is an adaptation process in individuals who experience long-term high-altitude exposure.

To balance the influence of the subjects’ emotions on their performance in the behavioral experiment, we adopted the Positive and Negative Affect Schedule (PANAS) and tested each participant before the behavioral experiment. Numerous studies have shown that emotions can alter an individual’s perception of time [40,41,42,43]. In research on the influence of current emotions on time perception, basic emotions including fear, anger, sadness, and happiness are included. In summary, different emotions can have different effects on time perception.

Our research shows that exposure to high altitude not only affects a series of basic cognitive functions such as attention and working memory [14,17,18] but can also have a negative impact on the time judgment abilities of migrants. When entering the plateau, with an increase in altitude, brain regions such as the prefrontal lobe, hippocampus, striatum, and cerebellum can be affected by hypoxia and exhibit significant functional decline, resulting in an impairment of cognitive functions [19,20]. In existing studies on the brain mechanisms of time perception, the cerebellum only has a specific timing function for a range of time intervals shorter than seconds [44]. The prefrontal cortex is mainly involved in the processing of time interval judgment over one second [21]. The hippocampus and striatum are involved in neural pathways in the brain regions that affect time information processing through neurotransmitters [32]. Taking these perspectives together, we can see similarities between time perception and the brain regions involved in previous studies on high-altitude exposure. When we enter high altitudes, these brain regions will be affected by the low pressure and low oxygen levels, resulting in a negative effect on time interval judgment.

At the same time, interval timing processing and time perception are related to arousal factors [21]. In Study 2, the direct effect of sleep-related factors (sleepiness or sleep deprivation) on time interval judgment ability (1080 ms) was marginally significant. However, when the intergroup factor of altitude (plateau vs. plain) was added, there was a significant interaction effect between altitude and sleepiness on time interval judgment (1080 ms), and altitude moderated the effect of sleepiness on time interval judgment (1080 ms).

Our research reveals that, at low sleepiness levels, the accuracy of time interval judgment in the high-altitude migrants was lower than that in the plain group, which may be due to the negative impact of the high-altitude environment on the time perception ability of the migrants. In a high-altitude environment, the brain regions closely associated with time interval judgment were affected by hypoxia, which resulted in a negative influence on individuals’ time interval judgment ability.

In particular, it was found that the time interval judgment accuracy of the migrant group (1080 ms) was significantly higher than that of the plain group under the condition of a high level of sleepiness. A long-term lack of oxygen supply can result in the sleep control areas of the brain becoming damaged. This can lead to changes in sleep structure and sleep quality, such as sleep rhythm disorders and sleep quality declines. Under the condition of a high level of sleepiness, the accuracy of time interval judgment in the high-altitude migrant group was higher than that in the plain group, which may be related to the poor sleep quality of the migrant group and their high sleepiness adaptation. Lack of sleep and a decline in sleep quality lead to an increase in individual sleepiness levels, which directly lead to the dysfunction of the hypothalamus–pituitary–adrenal axis, ultimately affecting the nutritional support of brain neurotrophic factors (including brain-derived and glial neurotrophic factors) to neurons and glial cells and resulting in cognitive impairment [45]. This can also lead to a decrease in blood flow to the prefrontal cortex, the region responsible for executive and behavioral functions. Cognitive function can also be impaired by affecting the prefrontal cortex [46]. Due to the increase in sleepiness levels caused by decreased sleep quality, these brain regions, which are highly connected to the processing of time interval perception, are negatively affected, finally negatively affecting the time interval judgment abilities of individuals.

In Study 3, altitude and sleepiness levels also had a significant interaction effect on time interval judgment abilities (1080 ms) under acute high-altitude exposure. With the increase in the level of sleepiness, the time interval judgment accuracy (1080 ms) of subjects in the acute exposure group decreased. At the same time, the performance of the acute exposure group was significantly lower than that of the plain group under both high and low levels of sleepiness. It can also be seen from Figure 7 that the time interval judgment performance of the acute exposure group exhibited a more severe downward trend with the increase in sleepiness level compared with that of the plain group. In Study 2, although there was no significant difference in the accuracy of time interval judgment between high and low sleepiness conditions in the long-term exposure group, the performance of the long-term exposure group showed an upward trend with the increase in the level of sleepiness. Here, time interval judgment performance under a high level of sleepiness was significantly better than that of the plain group. Study 3 confirmed the results of Study 2 in that individuals exposed to high altitudes long-term undergo an adaptation process. As individuals who had moved to the plateau adapted, their cognitive function recovered; however, it was still inferior to that of the plain residents.

The amount of time one lives at high altitudes is an important factor that affects the physiological adaptation to hypoxia. Zubieta-Calleja (2010) divides the adaptation process into three stages based on changes in human hematocrit: acute, subacute, and chronic. The formula for altitude adaptation with time and space is proposed as “high altitude adaptation factor = exposure time (day)/altitude (km)” [47]. For fixed elevations, the main factor is exposure time. The body undergoes a series of compensatory adaptation changes at high altitudes, which is called “acclimation”. According to certain physiological and physical indicators, Gao et al. (2001) divided the plateau acceleration into three stages: preliminary acceleration (more than 7 days on the plateau), basic acceleration (more than 1 month on the plateau), and complete occupation (more than half a year on the plateau) [48]. Physiology is the basis of psychological function, and different physiological stages of adaptation to high-altitude environments may affect cognitive function. It is generally believed that cognitive decline is most obvious when entering the plateau. With the extension of exposure time to high altitudes, cognitive function can recover to some extent after acclimation to high-altitude hypoxia; however, in the current study, participants still faced difficulty in reaching the same level as the plain group [19,49].

Previous studies have found that residence duration at high altitudes affects executive control function, which is significantly lower at 1 week and 2 years than at 1 month. Executive control function first declines, recovers, and then declines gradually with the extension of residence time [50]. In this study, the time interval judgment accuracy (1080 ms) of acute and long-term high-altitude exposure groups showed a completely different trend with the changes in sleepiness levels. We believe that the time interval judgment function of the settlers recovered and improved with the extension of exposure time; however, this was still not as good as that of the plain group. This is consistent with previous conclusions regarding the influence of exposure time to high altitude on individual cognitive function. In the future, we can conduct field follow-up studies on the same group of subjects to reveal changes in the settlers’ time judgment functions with the increase in their high-altitude exposure time.

There are several limitations in this study that future researchers should consider. First, the number of subjects used in this study was small. The next step will be to expand the sample size and more carefully control for additional variables such as gender and education level. Second, this study only considered the time distance judgment ability of high-altitude migrants; perhaps future studies should further investigate the time distance judgment ability of plateau dwellers. At the same time, the present study only investigated an altitude of 3650 m. At higher or extremely high altitudes, hypoxia may directly lead to obvious damage to time interval judgment, which may also be a direction for future study. In addition, this study only investigated the effects of the sleepiness dimension in circadian rhythm on time interval judgment. For low-pressure and low-oxygen plateau environments, the question of whether other specific factors affect time interval judgment is worth further study. Finally, it may be subjective to use a questionnaire method to evaluate the sleepiness or sleep deprivation levels of participants, which can be evaluated by objective evaluation indices such as polysomnography, in the future. We may also continue to use ERP, MRI, and other technologies to investigate the impact of high-altitude hypoxic environments on time interval judgment, which will help us to further explore the cognitive neural mechanisms of time perception injury in high-altitude hypoxic environments. At the same time, with the increase in the rates of car ownership and the number of drivers, the number of road traffic accidents has become high, and at present, human-related road traffic accidents account for the majority of the total number of accidents. Large amounts of statistical data and accident analysis results show that driver perception error is one of the main factors leading to traffic accidents [51]. Therefore, gaining an in-depth understanding of drivers’ perception, especially their perception of external space and collision time, would be of great significance for improving road traffic safety. This could help us to effectively prevent traffic accidents caused by perception errors and improve drivers’ driving behavior decisions and road facility design. This will also be a major direction for our future research, where we will carry out an ecological validity study to prove that high-altitude exposure affects individuals’ time perception and attempt to improve road traffic safety in high-altitude areas on this basis.

## 6. Conclusions

In conclusion, this study is the first of its kind to provide evidence that long-term high-altitude exposure can lead to the impairment of time interval judgment processing function, which suggests that high-altitude exposure has negative effects on time interval judgment. Based on our findings, we also found that sleep-related factors and high-altitude exposure interactively influenced time interval judgment. This study provides a theoretical basis for the high-altitude adaptation of workers in performing time perception tasks and extends the current research on high-altitude brain cognition. The new insight offered for workers entering the plateau is that they should not engage in work that requires fine and detailed cognitive processing within a week of their arrival. Allowing time for adequate rest and actively adapting to the plateau environment should be of the utmost priority. Finally, this study was conducted on migrants who grew up in plain areas and had entered high-altitude areas for the first time, staying for at least 1 year. This study may accurately reflect practical issues and is conducive to the promotion of migration to this region.

## Figures and Tables

**Figure 1 brainsci-12-00585-f001:**
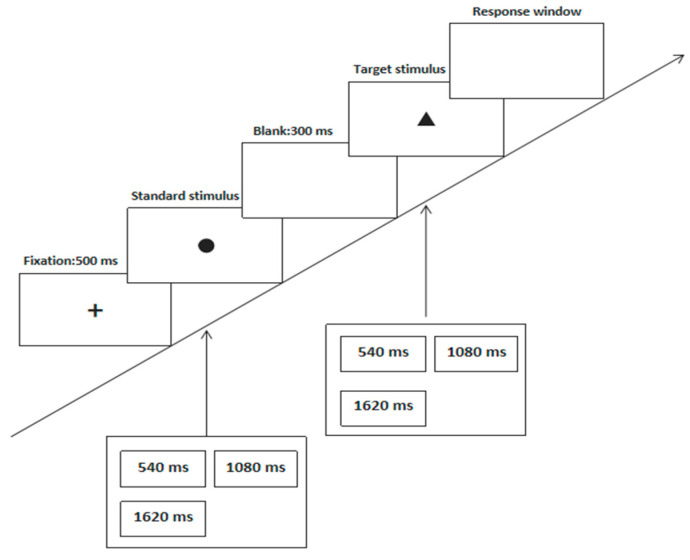
The experimental procedure of the “single stimulus method”.

**Figure 2 brainsci-12-00585-f002:**
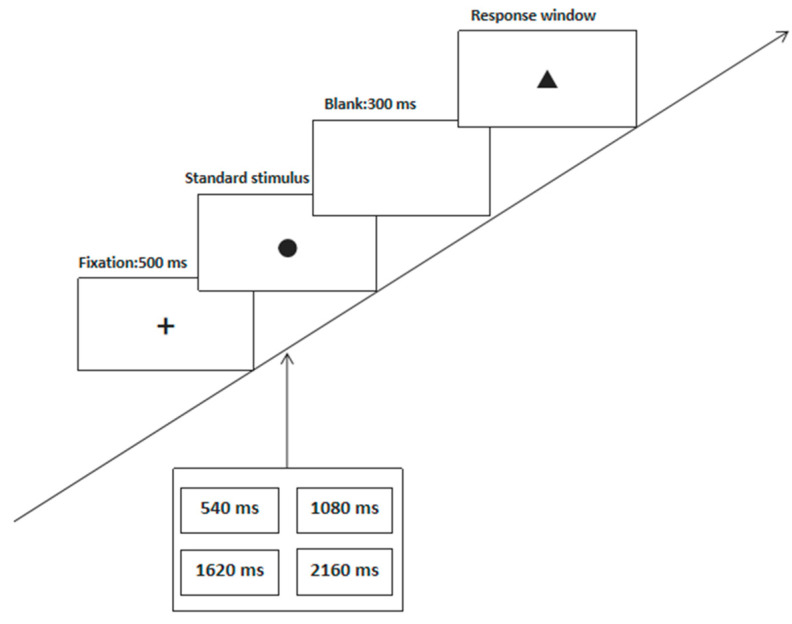
The experimental procedure of the “synchronization method”.

**Figure 3 brainsci-12-00585-f003:**
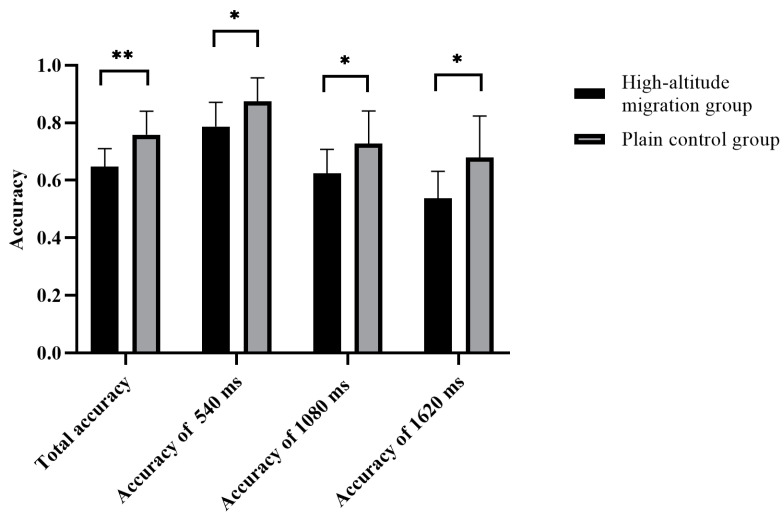
The difference in the accuracy rate of different groups in time interval judgment. * *p* ≤ 0.05; ** *p* ≤ 0.01.

**Figure 4 brainsci-12-00585-f004:**
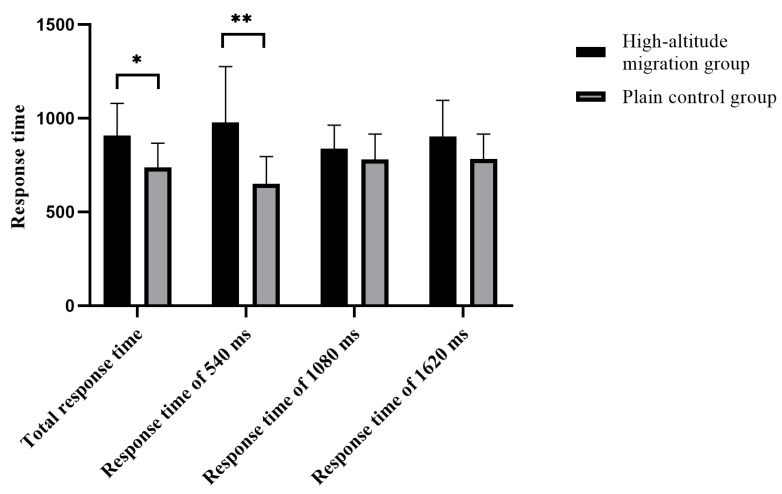
The difference in the response time of different groups in time interval judgment. * *p* ≤ 0.05; ** *p* ≤ 0.01.

**Figure 5 brainsci-12-00585-f005:**
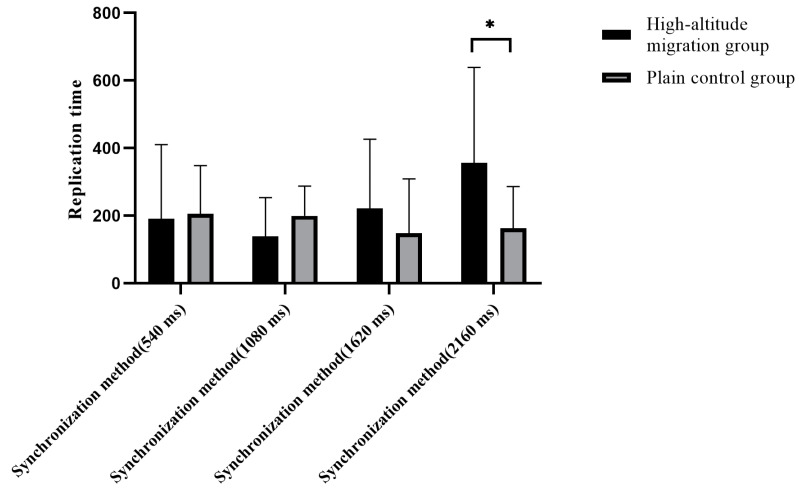
The replication time difference of the different groups for each stimulus in the synchronous stimulation method. * *p* ≤ 0.05.

**Figure 6 brainsci-12-00585-f006:**
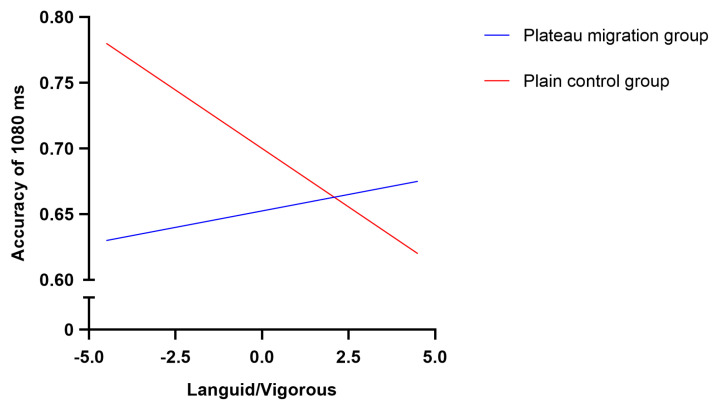
A simple slope test for the accuracy of time interval judgment (1080 ms) of languid/vigorous levels in the different groups.

**Figure 7 brainsci-12-00585-f007:**
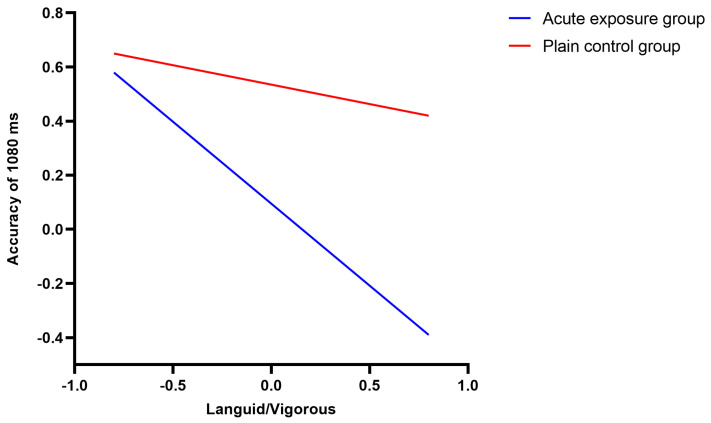
Difference in accuracy at 1080 ms between the acute exposure group and the plain control group.

**Table 1 brainsci-12-00585-t001:** Description of statistical and correlation analysis results.

	*M* ± *SD*	Languid/Vigorous	Accuracy of 1080 ms	Group
Languid/vigorous	19.04 ± 4.37	1		
Accuracy of 1080 ms	0.66 ± 0.11		1	
Group		0.38 *		1

* *p* ≤ 0.05.

**Table 2 brainsci-12-00585-t002:** A moderated mediation model of languid/vigorous levels for temporal accuracy (1080 ms).

Regression Equation		Fitting Index			Significance of Regression Coefficient		
Results of Variable	Predictor Variable	R	R^2^	F	*β*	*SE*	*t*
Accuracy of 1080 ms	Languid/vigorous	0.50	0.25	2.66	0.04	0.04	0.87
	Group				0.32	0.28	1.13
	Languid/vigorous * Group				−0.17	0.07	−2.58 *

* *p* ≤ 0.05.

**Table 3 brainsci-12-00585-t003:** The moderating effect of the plateau migration group and the plain control group on the accuracy of time interval judgment (1080 ms).

Adjust the Variable		Effect	BootSE	BootCI-Low	BootCI-Upper
Group	High-altitude migration group	0.11	0.07	−0.02	0.25
	Plain control group	−0.19	0.07	−0.33	−0.05

**Table 4 brainsci-12-00585-t004:** Descriptive statistics and correlation analysis results (N = 60).

	*M* ± *SD*	Languid/Vigorous	Accuracy of 1080 ms	Group
Languid/vigorous	18.80 ± 2.63	1		
Accuracy of 1080 ms	0.66 ± 0.11	−0.37 **	1	
Group		0.33 *		1

* *p* ≤ 0.05; ** *p* ≤ 0.01.

**Table 5 brainsci-12-00585-t005:** The moderated mediation effect of languid/vigorous level on the accuracy of time interval judgment (1080 ms).

Regression Equation		Fitting Index			Significance of Regression Coefficient		
Results of Variable	Predictor Variable	R	R^2^	F	*β*	*SE*	*t*
Accuracy of 1080 ms	Languid/vigorous	0.56	0.32	8.62	−0.59	0.13	−4.67 ***
	Group				0.46	0.16	2.83 **
	Languid/vigorous * Group				0.45	0.19	−2.33 *

* *p* ≤ 0.05; ** *p* ≤ 0.01; *** *p* ≤ 0.001.

**Table 6 brainsci-12-00585-t006:** The moderating effect of the acute exposure group and the plain control group on the accuracy of time interval judgment (1080 ms).

Adjust the Variable		Effect	BootSE	BootCI-Low	BootCI-Upper
Group	High-altitude migration group	−1.06	0.30	−1.67	−0.46
	Plain control group	−0.17	0.14	−0.45	0.10

## Data Availability

Data materials can be obtained by contacting the corresponding author.

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
