# Peer review of "High-Altitude Exposure and Time Interval Perception of Chinese Migrants in Tibet"

_brainsci, 2022, doi:10.3390/brainsci12050585_

Round 1
Reviewer 1 Report
The manuscript is very interesting and has a good theoretical and empirical foundation.
The main consideration is related to the structure and exposition of the data. The four studies have different subsections. It would be appropriate to try to homogenize the subsections.
Another aspect that should be taken into account is the fact that sometimes a subsection called "Results and discussion" appears. If the results are presented and at the same time a discussion of them is made, it is difficult to separate the original data from those with which they are discussed. For this reason, it is recommended that only Results be labeled. In the discussion, these results will be compared with previous results.
Thank you very much for your work.
Reviewer 2 Report
The present study is of interest to examine the effects of high-altitude exposure of Chinese migrants in Tibet, on their time interval judgment abilities based on three separate 3 studies:
Study 1
The first study investigated the effects of exposure to high-altitude and low-oxygen environments and the time interval judgment abilities of migrant settlers.
Study 2
In the second study, it was investigated whether prolonged exposure to a high-altitude environment had a greater negative impact on the time interval judgment abilities of sleep-deprived individuals.
Study 3
In the third study, it was investigated the interactive effects of acute high-altitude exposure and sleepiness on time interval judgment ability.
Despite the interesting work and upon sound study design, I strongly suggest following the comments to improve the quality of the manuscript. The manuscript is generally well written.
Abstract
- Please add some values (e.g, p and 95% CI values)
Methods
2. Please clarify how participants were really selected for each study. It was randomly?
3. Authors tested if age were significantly different between each compared group, for each study?
4. Please include the reference ethical number/code.
5. L 163-165: "The Cronbach’s α coefficient of the positive emotion subscale was 0.795, and the Cronbach’s α coefficient of the negative emotion subscale was 0.872." Please add a valid reference.
6. Figure 6: YY axis should start in 0 (e.g, include // between 0 and 0.60).
7. Authors should reinforce which type of inferential statistics were used for each study. As it stands it is not clear enough. In addition, statistical procedures might need to be discussed using a within-subjects approach since basic comparisons were performed. Given the high intra-individual variability, a within-subjects approach (recommended for small samples) might be appropriate (please see some recent works):
https://www.ncbi.nlm.nih.gov/pmc/articles/PMC2548822/, https://pubmed.ncbi.nlm.nih.gov/31527865/, https://pubmed.ncbi.nlm.nih.gov/33672683/; https://www.frontiersin.org/articles/10.3389/fphys.2021.678462/full).
For each study, please explain:
- Did all your results were normally distributed?
- How was this comparison attempted?
- Did the authors pool data at the individual level?
- How many data points were paired?
- Can repeated-measures ANOVA or t-test be used for such data set?
- Authors should also present 95% confidence interval
- I recommend presenting some figures (within-subjects), showing the main outcomes/results across the observation period, for each study.
8. I suggest the authors add some practical applications.
Round 2
Reviewer 2 Report
I am happy with the current version of the manuscript.
The authors did a good job on reviewing the manuscript and answering all the revisions maded.
I have only one highly recommendations for authors: Moderate English changes required
